# Determining the Impact of Heatwaves on Emergency Ambulance Calls in Queensland: A Retrospective Population-Based Study

**DOI:** 10.3390/ijerph20064875

**Published:** 2023-03-10

**Authors:** Hannah M. Mason, Jemma C. King, Amy E. Peden, Kerrianne Watt, Emma Bosley, Gerard Fitzgerald, John Nairn, Lauren Miller, Nicole Mandalios, Richard C. Franklin

**Affiliations:** 1Discipline of Public Health and Tropical Medicine, James Cook University, Townsville, QLD 4811, Australia; 2School of Population Health, Faculty of Medicine and Health, University of New South Wales, Sydney, NSW 2052, Australia; 3Information Support, Research & Evaluation, Queensland Ambulance Service, Brisbane, QLD 4031, Australia; 4School of Clinical Sciences, Faculty of Health, Queensland University of Technology, Brisbane, QLD 4000, Australia; 5School of Public Health and Social Work, Faculty of Health, Queensland University of Technology, Brisbane, QLD 4059, Australia; 6School of Biological Sciences, Faculty of Sciences, Engineering and Technology, The University of Adelaide, Adelaide, SA 5000, Australia; 7Disaster Management Branch, Queensland Health, Brisbane, QLD 4000, Australia

**Keywords:** heatwaves, climate change, ambulance, relative risk, excess heat factor

## Abstract

Heatwaves are a significant and growing threat to the health and well-being of the residents of Queensland, Australia. This threat is increasing due to climate change. Excess heat increases the demand for health services, including ambulance calls, and the purpose of this study was to explore this impact across Queensland. A state-wide retrospective analysis of heatwaves and emergency ‘Triple Zero’ (000) calls to Queensland Ambulance (QAS) from 2010–2019 was undertaken. Call data from the QAS and heatwave data from the Bureau of Meteorology were analysed using a case-crossover approach at the postcode level. Ambulance calls increased by 12.68% during heatwaves. The effect was greatest during low-severity heatwaves (22.16%), followed by severe (14.32%) and extreme heatwaves (1.16%). The impact varied by rurality, with those living in very remote areas and major cities most impacted, along with those of low and middle socioeconomic status during low and severe intensity heat events. Lag effects post-heatwave continued for at least 10 days. Heatwaves significantly increase ambulance call centre workload, so ambulance services must actively prepare resources and personnel to address increases in heatwave frequency, duration, and severity. Communities must be informed of the risks of heatwaves at all severities, particularly low severity, and the sustained risks in the days following a heat event.

## 1. Introduction

Heatwaves significantly impact human health and health systems [1,2,3]. Across Australia, increasing trends of record-breaking heat due to climate change are predicted over the next century, which will impact critical infrastructure and health service provision [1,2,4]. Heatwaves become disasters when communities and response agencies (e.g., ambulance services, emergency services, etc.) are unable to effectively respond or adapt to changing climate conditions [5,6]. Similarly, health services’ capacity to respond and adapt to service fluctuations caused by extreme heat is important for the health and well-being of individuals, communities, and regions. Effectively managing the impacts of heatwaves on health services, including ambulance services, requires an informed and prepared approach that is based on evidence [1,7,8].

Heatwaves impact health by increasing the risk of heat-stress-related conditions and exacerbating pre-existing conditions such as heart and renal disease [3,9,10]. For example, a recent study by Doan et al. using Queensland Ambulance Service (QAS) data found that heatwaves significantly increased the risk of out-of-hospital cardiac arrest in Brisbane by 1.19 to 1.48 times, depending on heatwave definition [11]. Ambulance services are often the first line of medical care during heatwaves, as individuals tend to stay within their homes to avoid heat exposure and may withhold from self-presenting to emergency departments [12]. Previous Australian research has found an increase in ambulance calls in the major cities of Sydney [13], Brisbane [8], Adelaide [9], Perth [14], and the state of Tasmania [15]. However, little is known regarding the state-wide impacts of heatwaves on demand for prehospital emergency services across the state of Queensland [16], a gap that our research addresses.

This study is part of the wider Heat, Health, and Human Environment Project, a collaboration between the Queensland Department of Environment and Science, Queensland Health, and Queensland Fire and Emergency Services under Strategic Objective 2 [17].

“*…Understand health system impacts, including morbidity and mortality, related to prolonged heat exposure and unseasonal heat, through data analysis at the state-wide level…*” (Strategic Objective 2) [17].

The purpose of the project is to improve understandings of heat-health relationships at a state-wide level [17], as the frequency, intensity, and duration of heatwaves continue to increase [1,18] (Figure 1, Appendix A). Under current projections, heatwaves in Queensland may last more than one month by the end of the century due to climate change [19]. In 2018, heatwave conditions persisted for 3% of the year, and this is expected to increase five-fold by 2090 (15%) without mitigation strategies [1]. Heatwaves are already impacting the health of Queenslanders [1,16], and the Queensland Ambulance Service (QAS) identified the need to quickly respond to the complex and compounding system-wide threats brought forth by climate change in their Strategic Plan (Strategic Objective 4, Sub-Objective 3) [20].

“*… To ensure the ongoing sustainability of ambulance and other health care services in Queensland in a constantly changing environment…*” (Strategic Objective 4).

“*…We will meet the threat of a changing climate through sustainable and socially responsible planning, resourcing and operations, and models of service tailored to the health care challenges posed by climate change…*”(Sub-Objective 3) [20].

In Queensland, emergency prehospital care is provided by the QAS in 15 geographic regions that align with the Queensland Health Hospital and Health Service Districts [21]. The QAS is a single, state-wide, government-funded emergency ambulance service that responds to over one million incidents annually [22]. The request for an emergency ambulance is made through a single national telephone number (Triple Zero [000]), and in Queensland, these calls are answered by Emergency Medical Dispatchers (EMDs) who work from seven Operations Centres located throughout the state [23]. If aeromedical retrieval is required, calls are directed to Retrieval Services Queensland (RSQ), a centralised coordination centre [24]. As well as emergency call-taking, operational deployment, and dispatch, these Operations Centres are responsible for the coordination of non-urgent patient transport services [22]. Calls for emergency ambulance service are triaged using the Medical Priority Dispatch System (MPDS), which is complemented by Computer-Aided Dispatch (CAD) to facilitate the rapid deployment of resources [23]. Most of these calls (approximately 78%) result in transportation to a medical facility for further treatment, such as public emergency departments, private hospitals, and other medical facilities [20].

The timely review of health service usage during heatwaves is a key element in developing effective health action plans [25]. The purpose of this study was to explore the impact of heatwaves on emergency ambulance calls from 2010 to 2019 by various factors such as rurality, socioeconomic status (SES), heatwave severity, time of day, and day of the week. Further, because the effects of heatwaves on people’s health and health service demand can extend for several days following a heatwave event (lag days) [13,21,26], a lag period of 10-days was examined to identify the length of effect on emergency ambulance calls following a heatwave event.

## 2. Materials and Methods

### 2.1. Study Setting

Queensland, colloquially referred to as the Sunshine State, is the second-largest state in Australia by size (1,730,648 km^2^), with more than half of the state’s landmass lying north of the Tropic of Capricorn [27]. The estimated resident population in 2021 was 5,240,520 persons, approximately 20% of the population of Australia [28,29]. The majority of the population is concentrated along the coast [30], but the state is also home to a geographically dispersed population further inland and along the expansive coast line (population density of less than 3 people per km^2^), which confers unique challenges in emergency prehospital management [21]. Heatwaves are becoming more frequent in Queensland, with increases in the number of heatwave days in recent years seen across the state (see Figure 2 and Figure 3).

### 2.2. Ambulance Call Data

CAD data were sourced from the QAS. Data on emergency (Triple Zero) ambulance calls were extracted for analysis, excluding the road component of aeromedical transfers, prebooked ambulance transport for scheduled appointments, return of patients from hospital to residence, and where an ambulance was prebooked to attend an event. Over a 10-year period from 1 January 2010 to 31 December 2019, there were a total of 6,846,589 calls over 1,629,015 total days (365.25 days × 446 postcodes × 10 years). CAD data were collected from the Advanced Medical Priority Dispatch Software for the purposes of resource allocation and dispatch. Extracted data included date of incident (day, time), location of incident (by suburb and postcode), dispatch priority, and MPDS code. MPDS code is a response determinant assigned at point of call-taking that identifies the nature and time criticality of each incident [32].

### 2.3. Climate Data

Climatic information was obtained from the Bureau of Meteorology (BOM) from 1 January 2010 to 31 December 2019 by Statistical Areas Level 2 (SA2) region. SA2 regions were converted to postcodes using weights according to the Postcode 2016 to Statistical Area Level 2 2016 Australian Bureau of Statistics (ABS) concordance file https://abs.gov.au; 1270.0.55.001 (accessed 15 March 2022). These files match postcodes with suburb information to enable data integrity of geocoding information. For postcodes relating to SA2 regions in areas of Queensland that border other states, corresponding weights were redistributed across the SA2 regions in Queensland only (postcodes 4383, 4385, 4493, 4824).

Heatwave data were presented as Excess Heat Factor (EHF), a metric developed by Nairn and Fawcett [33]. The EHF is location-dependent and accounts for short-term and climate-scale temperature anomalies [34], but does not explicitly include humidity information [35]. Heatwave days were classified as days in which the EHF > 0 and were further categorized into low-severity heatwaves (EHF value >0 to <1), severe heatwaves (EHF value 1 to <3) and extreme heatwaves (EHF value 3+). Where ‘heatwave days’ are discussed, this includes all days of all severities; otherwise, the specific severity level is stated.

### 2.4. Analysis

Data analyses were conducted using IBM SPSS statistics version 28 and Microsoft Excel 2016. A retrospective case-crossover approach was undertaken by deriving matched data. To generate the matched sample, cases that occurred within a postcode on a day of the year (1–366) that was a heatwave day at least once across the 10-year period were included. For example, if there was a heatwave at least once on day 18 (18 January) during the study period in postcode 4814, and there was an emergency ambulance call at least once on day 18 in the same location, this was considered a ‘matched’ day and was included in the analysis. We noted that there were minimal differences in emergency ambulance calls across days of the week for non-heatwave versus heatwave days (Figure 4).

Using the matched data, the number of ambulance calls that occurred on heatwave days was compared with the number of ambulance calls that occurred on the same day of the calendar year (postcode specific) in which a heatwave did not occur. A similar process was utilized to explore lag-days, where cases were separately matched to each lag-day for analysis. Lag-day matching was completed for 10 days following each heatwave, which is a lag period used in previous epidemiological heatwave research [36,37,38]. Using Excel, the first non-heatwave day following a heatwave event (where the EHF was <0) was designated as lag-day 1, the second day was designated as lag-day 2, and so on until lag-day 10. The designation of lag-days was stopped if another heatwave occurred within the 10 days following the initial event. Therefore, there were no lag-days that were heatwave days.

Postcodes were categorized into five rurality groupings determined by the Australian Statistical Geography Standard Remoteness Structure (ASGS) as major cities, inner regional, outer regional, remote, or very remote [39]. The Index of Relative Socio-economic Advantage and Disadvantage (IRSAD) was used as a measure of socioeconomic status [40]. IRSAD provides information regarding the socioeconomic conditions within an area based on Census data and includes 25 variables such as household income, the number of people in skilled occupations, education, the number of one-parent families, and access to the internet [40]. Using the ABS Postal Area, SEIFA, 2016 concordance file, each postcode was matched to a corresponding IRSAD decile (10% grouping; https://www.abs.gov.au/AUSSTATS/abs@.nsf/DetailsPage/2033.0.55.0012016?OpenDocument; accessed 15 March 2022). Postcodes were grouped into low SES (most disadvantaged; deciles 1–3), medium SES (deciles 4–7), and high SES (most advantaged; deciles 8–10).

Relative risk and 95% confidence intervals for ambulance calls on heatwave days versus non-heatwave days were calculated using the below formula. Dates were converted into financial years (1 July–30 June) to capture the summer months in Queensland within each year period (summer months span December–February). For each financial year in the study period, relative risk was calculated by EHF severity groupings. Relative risk was also calculated by rurality and socioeconomic groupings stratified by severity, and separately for each lag-day (1–10).
$$Relative\;risk=\frac{number\;of\;calls\;on\;heatwave\;days\;÷number\;of\;heatwave\;days}{number\;of\;calls\;on\;non\;heatwave\;days÷number\;ofnon\;heatwave\;days}$$

To visually examine differences in ambulance call demand on heatwave versus non-heatwave days by the time of day and day of the week, proportional line graphs were created. Further, chi-squared analyses were run to explore whether the differences were statistically significant. Cases were then categorised into broad “reason for seeking attendance” according to MPDS code, assigned by dispatchers at point of call-taking (Appendix A). To explore changes in demand on heatwave vs. non-heatwaves days across dispatch codes, chi-squared analyses were performed. Bonferroni corrected pairwise z-tests were conducted to determine the specific nature of the differences. The significance level (α) was set to 0.05.

Heatwave days of all severities were mapped across postcodes using ArcGIS Pro software, with spatial data derived from the ABS Australia-wide ASGS Edition 2016 shapefile https://abs.gov.au; 1270.0.55.001 (accessed 8 November 2022).

Ethics approval was granted by the Children’s Health Queensland Hospital and Health Service Human Research Ethics Committee (HREC/21/QCHQ/72044). Following ethics approval, approval was sought for the accessing the ambulance data from QAS (QAS114).

## 3. Results

Over the 10-year study period from 01 January 2010 to 31 December 2019, pooled across all matched postcodes (n = 446), there were 143,767 heatwave days and 383,949 non-heatwave days. Across heatwave days, there were 740,405 emergency ambulance calls, and across non-heatwave days, there were 1,754,864 emergency ambulance calls. Overall, the relative risk of emergency ambulance calls was 12.68% greater during heatwave days than non-heatwave days (RR: 1.13, CI: 1.12–1.13).

Heatwave severity impacted the rate of emergency ambulance calls across the study period. There were 52,892 low-severity heatwave days (10.02% of total days), 41,415 severe heatwave days (7.85% of total days), and 49,460 extreme heatwave days (9.37% of total days) across Queensland postcodes (n = 446). The rate of ambulance calls on heatwaves days vs. non-heatwave days was 22.16% higher for low-severity heatwaves (RR: 1.22, CI: 1.21–1.23), and 14.32% higher for severe heatwaves (RR: 1.14, CI: 1.13–1.16). During extreme heatwaves, the rate of ambulance calls was 1.16% higher on heatwave vs. non-heatwave days (RR: 1.01, CI: 1.00–1.02). The impact of heatwave severity on ambulance calls varied across financial years, with no clear patterns established (Figure 5).

The risk of ambulance calls was significantly higher on heatwave days vs. non-heatwave days for all rurality groupings, but most pronounced in very remote areas (RR: 1.16, CI: 1.13–1.20), followed by major cities (RR: 1.12, CI: 1.11–1.12), then inner regional (RR: 1.09, CI: 1.08–1.11), remote (RR:1.05, CI: 1.02–1.08) and outer regional areas (RR: 1.04, CI: 1.02, 1.05).

The effect of heatwave severity on the risk of emergency ambulance calls differed by rurality grouping (Table 1). The rate of calls was significantly greater during low-severity heatwaves in comparison to non-heatwave days for major cities, inner regional, outer regional, and very remote areas, but was most pronounced in very remote areas (45.15% greater on low-severity heatwave days versus non-heatwave days). For severe heatwaves, the rate of ambulance calls was significantly higher in all remoteness categories in comparison to non-heatwave days, with the effect being greatest in very remote areas (27.48% greater on severe heatwave days versus non-heatwave days). During extreme heatwaves, the rate of ambulance calls was significantly higher in comparison to non-heatwave days for major cities (12.34%), but the rate was significantly lower in inner regional, outer regional, and very remote areas.

Overall, rates of ambulance calls were higher for all IRSAD categories (low, medium, and high) during heatwave vs. non-heatwave periods; however, this varied by heatwave severity (Table 1). Call rates were higher on heatwave vs. non-heatwave days during low severity and severe heatwaves for each IRSAD category, but this was most pronounced in areas categorised as low and medium IRSAD, during low-intensity heatwaves. During extreme heatwaves, there were fewer calls on heatwave vs. non-heatwave days in areas categorised as low and medium IRSAD, but more calls in areas categorised as high IRSAD.

The proportion of ambulance calls by the hour of the day showed a similar pattern for heatwave and non-heatwave days (Appendix A). The proportion of ambulance calls by weekday also showed a similar pattern for heatwave and non-heatwave days, with greater fluctuation noted across the week on heatwave days (Figure 2; Appendix A).

The proportion of ambulance calls by dispatch code varied across heatwave and non-heatwave days (Appendix A). There was a higher proportion of calls on heatwave days for heat exposure, mental health, other/transport, and specified medical; however, there was a higher proportion of calls on non-heatwave days for cardiac, injuries, obstetric, and respiratory MPDS codes. There were no significant differences in ambulance calls relating to cold exposure or stroke.

The lag-day analysis showed significant increases in emergency ambulance calls for each of the 10 days following heatwave events (Figure 6). The rate of ambulance calls was highest on the second and third days and lowest on the fifth and sixth lag-days.

## 4. Discussion

In this study of emergency ambulance calls (Triple Zero) in Queensland over a ten-year period, the rate of calls was significantly higher during heatwaves (12.68%) than on non-heatwave days, and significant increases occurred for at least ten days post-heatwaves (lag). This effect of ambulance calls increasing during heatwaves varied by severity, rurality, and socioeconomic status. The increase in calls is similar to the results of other Australian studies, including a reported 14% increase in ambulance calls during a Sydney heatwave in 2011 [13], a 10% and 16% increase in ambulance callouts (excluding ambulance transport) during the 2008 and 2009 Adelaide heatwaves [9], respectively, and an 18% increase in ambulance attendances in the capital city of Brisbane during heatwaves from 2007–2011 [8].

### 4.1. Impacts on Ambulance Services

Increased ambulance calls (12.04–13.32%), during heatwave periods impact resourcing and operations and pose real challenges for emergency services [41]. This increase in workload follows the normal ambulance call patterns by time of day and day of week, and as such, has implications for staff, infrastructure, and modality resourcing. This has the potential to compound or expose existing system frailties.

Heat stress may cause system failures when infrastructure at dispatch centres reaches thermal limits, causing the malfunctioning of computers, ICT, and air-conditioning, reducing operational efficiency, so this must be mitigated through system design [41]. Staff are at risk, as heatwaves increase heat-related illness for workers as well as the likelihood of other workplace injuries (wounds, burns, etc.) [41]. Furthermore, extreme heat can compromise supply chains and impact the delivery of necessary supplies [41]. As Queensland already faces increasing temperatures and heatwaves, a systems-thinking approach to adaptation strategies is required to proactively tackle the challenges ahead [41].

Research measuring the impact of climate conditions on emergency health services has been identified as a strategic priority for QAS to inform future planning and response models that are tailored to the healthcare challenges posed by climate change [20]. The research presented here can be used by ambulance services more broadly to ensure that services are adequately prepared and have appropriate strategies in place to meet the threat of a changing climate. While this study explored Triple Zero calls specifically, it is assumed that increases in call demand would also have flow-on effects regarding ambulance attendance, emergency department attendance, and potentially through to hospital admissions, and this will be explored in further work under the Heat, Health, and Human Environment (HHHE) project.

There is a need for emergency health services, including ambulance services, to engage with the community around public health messaging and associated risk mitigation in order to reduce the impact of heatwaves. Though this study found that Triple Zero calls for heat exposure, mental health, other/transport, and specified medical conditions increased during heatwaves, further investigations of actual reasons for ambulance attendance as recorded by attending paramedics (vs reason for call) are required and will be conducted as part of the HHHE project. Furthermore, demographic differences in heatwave risk including age, sex, and Aboriginal and Torres Strait Islander status will be explored. Identifying those at greatest risk will help to mitigate the impacts of increasing heatwave activity on limited emergency health resources to reserve limited emergency health resources and protect the health and safety of the community.

### 4.2. Heatwave Severity

There was an inverse association between the relative risk of emergency ambulance calls and heatwave severity. The greatest increase in calls for heatwave versus non-heatwave days occurred during low-severity heatwaves (22.16%), followed by severe heatwaves (14.32%), and then extreme heatwaves (1.16%). This finding may be attributed to risk perception. That is, it is possible that Queenslanders are aware of the dangers of extreme heatwaves and mitigate risk by staying indoors but are more willing to risk exposure during low-severity heatwaves. However, adaptive behaviours have not yet been explored in Queensland, and further research must be done to better understand heatwave severity and risk perception.

The inverse relationship found in this study differs from results found in Tasmania, where ambulance attendance over a 12-year period increased with heatwave EHF severity (4% increase for low-intensity heatwaves, 10% increase for severe heatwaves, and 34% for extreme heatwaves) [15]. However, this study was based on ambulance attendance data (versus ambulance call data). A direct relationship between ambulance callouts and heatwave severity was also found in South Australia; however, there was some regional variability [42]. The differences between the present findings and the findings from Tasmania and South Australia may be due to the already high temperatures found in Queensland during the warmer months of the year. For example, a low-intensity heatwave is hotter in Queensland than in Tasmania and South Australia, as the EHF is relative to local climate [33]. Further work needs to be undertaken to explore the impact of low-severity heatwaves on ambulance services, including Australia-wide comparisons.

### 4.3. Rurality and Socioeconomic Status

Demonstrating the areas of greatest vulnerability across the state can help ambulance services to develop targeted mitigation strategies such as developing early warning systems and targeted public safety messaging [43]. This study identified that the highest increase in ambulance calls on heatwave versus non-heatwave days occurred in very remote and metropolitan areas. This finding differed by heatwave severity. For major cities, increases in ambulance calls were similar across all heatwave severities, but there was an inverse association between relative risk of ambulance calls and heatwave severity in inner regional, outer regional, and very remote areas. These findings differ from a population-based study in New South Wales, which found a dose–response relationship between EHF heatwave severity and ambulance call-outs for each level of remoteness [44]. Emergency ambulance calls (Triple Zero) are coordinated through eight Operation Centres in Queensland [45], and therefore the results of this study can be used to inform the distribution of resources and staff during heatwave events according to the location of each centre. The results from the present study also indicate that heatwave messaging should be tailored by rurality. Metropolitan dwellers must be aware of the consistent risk during all heatwaves, and those living outside major cities must be aware of the health risks during low-severity heatwaves.

There were minimal differences regarding socioeconomic status, as measured by IRSAD, and heatwaves overall. The relative risk of ambulance calls ranged from 1.11 (95% CI: 1.10, 1.12) for the most disadvantaged group to 1.13 (95% CI: 1.12, 1.14) for the most advantaged group. This is likely due to the use of postcodes when presenting SES, which may not provide enough fidelity for the exploration of differences by SES. The importance of SES as an indicator of heatwave risk has been highlighted in the literature [43]. In a recent Australia-wide literature review, it was determined that low socioeconomic status was one of the common risk factors for heatwave presentations to health services and mortality [16]. Socially disadvantaged groups are often associated with poorer general health, limited social support, limited access to health services, and lifestyle risk factors influencing their vulnerability to extreme heat [46]. Further explorations of the risk of heatwaves by socioeconomic status using a more precise indicator for social disadvantage in Queensland are required.

### 4.4. Lag Effects

In the present study, a significant lag effect for ambulance calls was found across 10 lag-days (Figure 6). Lag effects of heatwaves on health service utilisation have been indicated in previous research in Australia [43]. For example, a multi-city study found that the lag effects of heatwave mortality were significant up to the fifth day following a heatwave in Brisbane [47]. It is possible that the lag effect found in this study may be an artefact of high-temperature days following a heatwave event that do not meet the heatwave threshold but are still elevated to the point of having a physiological impact. High ambient temperatures independent of heatwaves have been shown to increase emergency ambulance attendances in Brisbane by 1.17% for each 1 °C increase above 22 °C (0–1 days lag) [48]. These findings highlight the need for community and health system awareness of the continuing risk of health consequences in the days following a heatwave event. Furthermore, the ambulance services must prepare for the lag impacts of heatwaves on call demand and appropriately tailor staff rostering and resourcing.

### 4.5. Other Factors Impacting Heatwaves and Health

Air quality is an important consideration for the impact of heatwaves on health. In Australia, air pollution has been indicated as a significant risk factor for ambulance callouts in Perth [14], but this relationship has not been explored elsewhere in Australia [16]. Conducting a statewide analysis to determine the impact of air quality on heat-health relationships is complicated by the positioning of air quality stations in Queensland. Currently, air quality stations are concentrated around the coast, with one station located in the northwest quadrant of the state, and no stations in the southwest quadrant [49]. Therefore, air quality was not included in this analysis but is suggested for community-level work where monitoring stations are positioned.

Individual level risk factors including socioeconomic status [43,50], low vegetation [51,52], air conditioning [53], social supports [50,53,54], urban dwelling [44], and coastal dwelling [14] have been shown to impact heatwave health service demand in Australia. This information cannot be determined by the data in this study, and would require individual prospective monitoring, which may be valuable, though extremely challenging when taking a state-wide approach.

### 4.6. Strengths and Limitations

This study had several strengths. It is the first Queensland-based study to undertake a statewide approach to explore the impacts of heatwaves on emergency ambulance calls. Using population-based data, ambulance call demand was explored across a 10-year period. The ability to utilise ambulance call and dispatch data is a low-cost mechanism that provides insights into individuals’ perceived need for urgent medical care [55]. Ambulance call data were matched to Excess Heat Factor (EHF) data, which is considered a better predictor of health service utilization in comparison to the use of daily average temperature, three daily average temperatures, and three daily maximum temperatures [7].

This study also had limitations. Potential climatic confounding variables including air pollution and humidity were not explored. High vapour pressure can increase heat stress, as it makes it difficult for sweat to evaporate and the body to cool down [35,56]. The Excess Heat Factor can be used to provide effective heatwave guidance for most climatic regions in Australia but may not provide competent early warning guidance for rare very dry or very humid heatwaves in the tropics [35]. Additionally, a synergistic effect exists between air pollutants and high temperature on health outcomes, which was not adjusted for in this study [57]. Recent studies have shown that exposure to extreme heat and air pollution has a larger effect on mortality beyond the sum of their individual effects [58]. Analyzing air pollution data across the state of Queensland is challenging, as it comes with several limitations and confounding variables [59]; this was outside of the scope of this project.

This study used postcodes as the geographic discriminator; however, there is large geographic, climatic, and demographic variability (e.g., SES), specifically in rural regions, which limits the ability to draw specific conclusions within some postcodes. Further, socioeconomic status was approximated by postcode using the 2016 ASGS correspondence file. An individual’s true socioeconomic status may not be adequately reflected by the average socioeconomic status of their postcode, and there is potentially large variability within each postcode. Thus, these results should be interpreted with caution.

Differences in ambulance calls were analysed by dispatch code. Dispatch codes are based on the details provided by the caller to the dispatcher and may not accurately reflect the reason for the call, ambulance attendance, or final medical diagnoses. Hence, the associations reported by MPDS should be interpreted with caution. Additionally, call-taking data may not accurately reflect resource consumption. For example, the number of calls to 000 may not equate with the number of ambulance attendances, as some incidents may have multiple patients, some may be referred to an alternate response other than receiving an emergency ambulance response (e.g., aeromedical retrieval), and some incidents may receive multiple ambulance units in response. These data also do not reflect the total volume of work, i.e., how long the dispatcher spends on each call, which may also change during a heatwave. Future research using ambulance attendance data and other health service data (emergency department attendance, hospitalisation, and linked data) would usefully inform understanding of the nature and magnitude of health complaints that increase due to heatwaves.

## 5. Conclusions

Heatwaves significantly (12.68%) increase ambulance call demand in Queensland, especially during low-severity events, and continuing post-event for at least 10 lag-days. These impacts are significant across all geographic groupings by rurality and socioeconomic groups, with the greatest increases seen in very remote areas and major cities. Alongside predicted increases in heatwave duration, intensity, and frequency across the state due to the effects of climate change, the results of this study indicate a need for tailored community messaging and targeted resourcing for ambulance services surrounding heatwaves. The implications for future climate forecasts that predict limited reprieve between heatwave events suggest health system strengthening is a worthwhile investment. It is imperative for ambulance services to adequately prepare for future heatwaves, including increased calls to Triple Zero.

## Figures and Tables

**Figure 1 ijerph-20-04875-f001:**
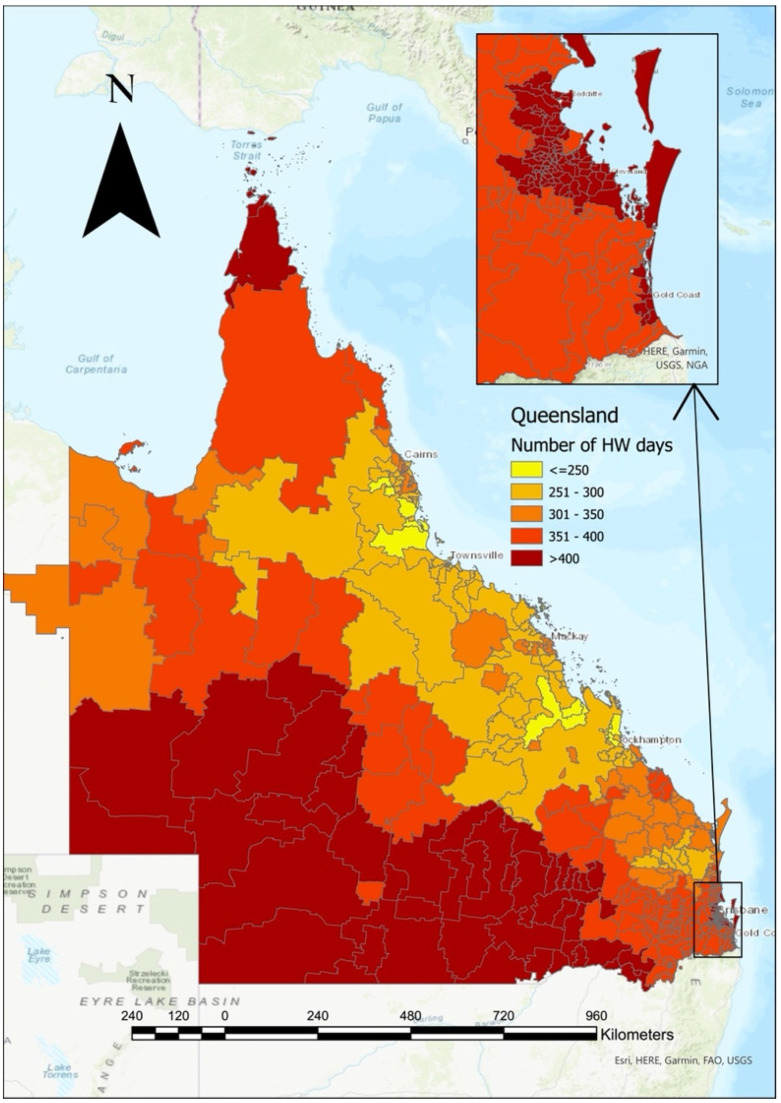
Number of heatwave (HW) days per postcode (Queensland; 2010–2019). Note: See Appendix A for full-size version.

**Figure 2 ijerph-20-04875-f002:**
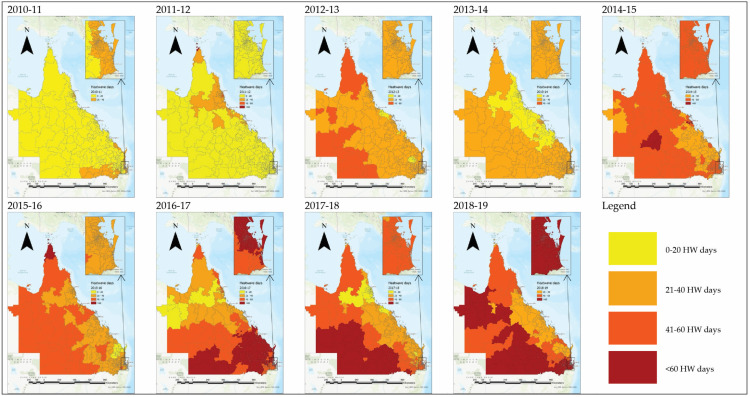
Heatwave (HW) days per postcode across Queensland for the 2010/11–2018/19 financial years. Note: La Niña occurred during the 2010–12 period, and El Niño occurred during the 2015–16 period [31]. Heatwaves are likely to be much longer and more frequent during El Niño in comparison to La Niña in Queensland [19]. Where La Niña typically brings above average rain in Australia, El Niño brings drought [1]. See Appendix A for full-size version.

**Figure 3 ijerph-20-04875-f003:**
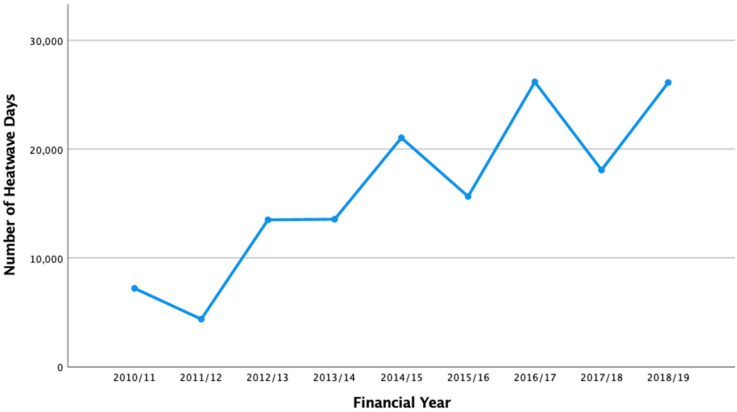
Trends over time for number of heatwave days (Queensland; 2010/11–2018/19). Note: For this study, 446 postcodes were included across Queensland; as such, there are in total 1,629,015 possible days across the 10-year period (162,901.5 per financial year).

**Figure 4 ijerph-20-04875-f004:**
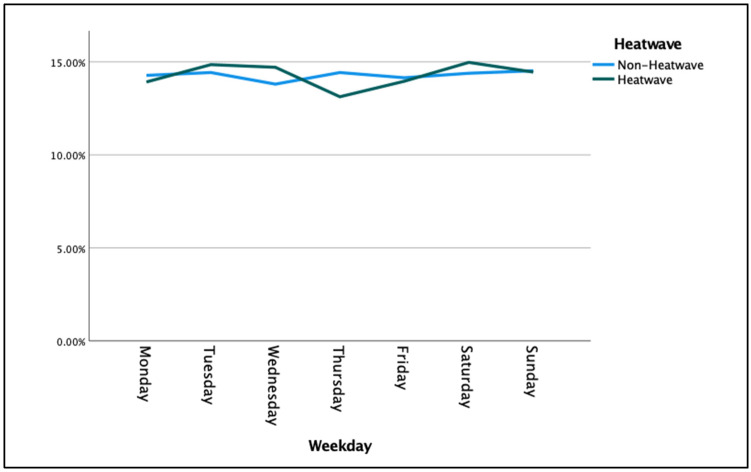
Proportion of ambulance calls for heatwaves vs. non–heatwaves by day of the week (Queensland; 1 January 2010 to 31 December 2019).

**Figure 5 ijerph-20-04875-f005:**
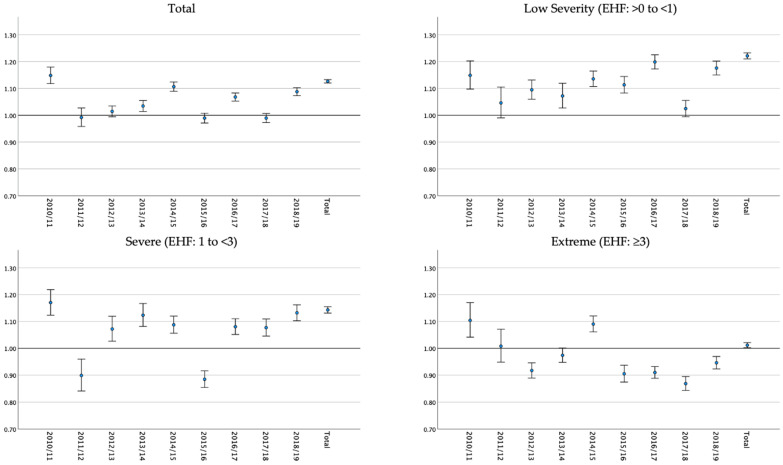
Relative risk of emergency ambulance calls on heatwave vs. non-heatwave days by financial year for total, low, severe, and extreme heatwaves.

**Figure 6 ijerph-20-04875-f006:**
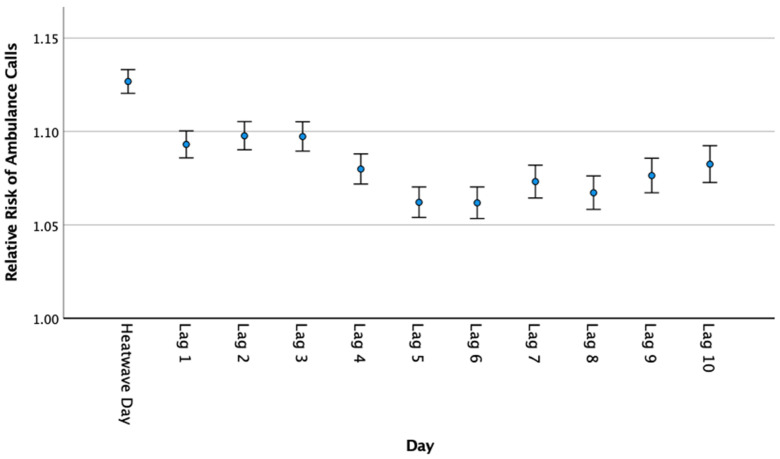
Relative risk of ambulance calls on days following a heatwave (lag-days; Queensland; 2010–19). Note: If a heatwave fell within the 10 lag-days, the designation of lag-days was stopped.

**Table 1 ijerph-20-04875-t001:** Relative risk of ambulance calls on heatwave vs. non-heatwave days by rurality, socioeconomic status, and heatwave severity (Queensland; 2010–19).

Relative Risk (95% Confidence Interval)					
		Low	Severe	Extreme	Total
Rurality	Major Cities (n = 1,559,021)	1.11 (1.10, 1.13) *	1.11 (1.09, 1.12) *	1.12 (1.11, 1.14) *	1.12 (1.11, 1.12) *
	Inner Regional (n = 499,232)	1.30 (1.27, 1.33) *	1.10 (1.07, 1.13) *	0.94 (0.92, 0.96) *	1.09 (1.08, 1.11) *
	Outer Regional (n = 372,867)	1.27 (1.25, 1.30) *	1.04 (1.02, 1.07) *	0.77 (0.76, 0.79) *	1.04 (1.03, 1.06) *
	Remote (n = 786,301)	0.95 (0.90, 1.00)	1.18 (1.12, 1.25) *	1.02 (0.98, 1.07)	1.05 (1.02, 1.08) *
	Very Remote (n = 22,265)	1.45 (1.38, 1.52) *	1.27 (1.20, 1.35) *	0.84 (0.80, 0.88) *	1.16 (1.13, 1.20) *
IRSAD	High (n = 613,194)	1.11 (1.09, 1.13) *	1.12 (1.10, 1.14) *	1.09 (1.07, 1.11) *	1.13 (1.12, 1.14) *
	Middle (n = 1,090,899)	1.25 (1.23, 1.27) *	1.14 (1.12, 1.16) *	0.98 (0.97, 1.00) *	1.12 (1.11, 1.14) *
	Low (n = 786,301)	1.27 (1.25, 1.29) *	1.12 (1.10, 1.14) *	0.98 (0.97, 1.00) *	1.11 (1.10, 1.12) *

* Denotes statistical significance; n = number of calls.

## Data Availability

Data are available via Queensland Ambulance Service once ethical approval for the use of the data has been granted.

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
