# Peer review of "Determining the Impact of Heatwaves on Emergency Ambulance Calls in Queensland: A Retrospective Population-Based Study"

_ijerph, 2023, doi:10.3390/ijerph20064875_

Round 1

Reviewer 1 Report

It is a manuscript whose objective is. "Determining the Impact of Heatwaves on Emergency Ambulance Calls in Queensland: A Retrospective Population-based Study".

The introduction does not reflect the current state of knowledge on the subject. There are publications similar to the one presented here that are not included in the work. Authors should reference similar papers.

Methodologically the authors should explain how the possible effect of air pollution and noise is considered in their study. The effect of chemical air pollution on heat waves can have a greater effect than the heat wave itself. This aspect should be discussed in depth.

The rural or urban character modifies the impact that heat waves have on morbidity and mortality. Is the definition of heat wave the same for rural populations as for urban populations? What role do other factors such as the existence of air conditioning, the age of the buildings, the existence of green areas, the thermal island effect, the level of income or the age of the population play in this investigation?

They are key factors that can explain the results obtained and that must be discussed and analyzed.

Author Response

Thank you for providing us with the opportunity to respond to the reviewer’s comments. Please see our responses below along with the corresponding manuscript.

Reviewer 1:

Reviewer Comment

Response

The introduction does not reflect the current state of knowledge on the subject. There are publications similar to the one presented here that are not included in the work. Authors should reference similar papers.

Thank you. We have added more references in the introduction (see: Campbell et al., 2021; Doan et al., 2022; Tong et al., 2019; and Patel et al. 2019). We recently published comprehensive literature review on this topic (https://doi.org/10.1186/s12913-022-08341-3).

Methodologically the authors should explain how the possible effect of air pollution and noise is considered in their study. The effect of chemical air pollution on heat waves can have a greater effect than the heat wave itself. This aspect should be discussed in depth.

Thank you, we have discussed this further in the limitations.

The rural or urban character modifies the impact that heat waves have on morbidity and mortality. Is the definition of heat wave the same for rural populations as for urban populations?

The definition used for heatwaves by the Bureau of Meteorology, the Excess Heat Factor (as noted in the methods), is the same across Australia regardless of rurality.

What role do other factors such as the existence of air conditioning, the age of the buildings, the existence of green areas, the thermal island effect, the level of income or the age of the population play in this investigation?

Thank you for this comment. These are great questions but are outside of the scope of this paper. These are areas in which we hope to follow up in the future. While this study looked at a macro-level, we agree that individual level factors are important and need to be explored in more detail.

Reviewer 2 Report

The article is well written. The introduction is sufficiently detailed and provides a good overview of the state of art. Chapter 2 - "Materials and Methods" is understandable. Here, however, it is necessary to prove that the number of emergency calls does not change significantly during the week. If the day of the week was significant, it would not be possible to compare January 18 in individual years. A graph showing this is provided in "Supplementary Figure 13". Insert it in this section, describe and prove that the day of the week has no significant influence. The presented results are interesting, clear, relevant, and useful. In the article, a 95% confidence interval is used to express the results. The discussion is generally processed in a comprehensible way and will focus on individual results and limitations. I consider the lag-day analysis to be very suitable. The description of data processing for lag-days could be expanded.
I consider the conclusions correct and valuable.

Author Response

Thank you for providing us with the opportunity to respond to the reviewer’s comments. Please see our responses below along with the corresponding manuscript.

Reviewer 2:

Reviewer Comment

Response

The article is well written. The introduction is sufficiently detailed and provides a good overview of the state of art.

Thank you for your time and feedback.

Chapter 2 - "Materials and Methods" is understandable. Here, however, it is necessary to prove that the number of emergency calls does not change significantly during the week. If the day of the week was significant, it would not be possible to compare January 18 in individual years. A graph showing this is provided in "Supplementary Figure 13". Insert it in this section, describe and prove that the day of the week has no significant influence.

Thank you. We have moved Supplementary Figure 13 into the methods section (see Figure 2).

The presented results are interesting, clear, relevant, and useful. In the article, a 95% confidence interval is used to express the results. The discussion is generally processed in a comprehensible way and will focus on individual results and limitations. I consider the lag-day analysis to be very suitable. The description of data processing for lag-days could be expanded.

Thank you. We found the lag-analysis very interesting as well and look forward to exploring it in greater detail in further studies. We have added detail regarding the processing of lag-days.

I consider the conclusions correct and valuable.

Thank you.

Round 2

Reviewer 1 Report

Questions 2 and 4 have not been adequately answered.

It should be included what it means both quantitatively and qualitatively not to include noise and pollution in the analysis. This is not to be put as a mere limitation of the study.

The same applies to the questions raised in point 4.

Author Response

Dear Reviewer

Thank you for your comments of the paper.  We have added a new section in the discussion to address you comments.

Re: Determining the Impact of Heatwaves on Emergency Ambulance Calls in Queensland: A Retrospective Population-based Study

Thank you for providing us with the opportunity to respond to the reviewer’s comments. Please see our responses below along with the corresponding manuscript.

Reviewer 1:

Reviewer Comment

Response

Question 2: Methodologically the authors should explain how the possible effect of air pollution and noise is considered in their study. The effect of chemical air pollution on heat waves can have a greater effect than the heat wave itself. This aspect should be discussed in depth.

Question 4: What role do other factors such as the existence of air conditioning, the age of the buildings, the existence of green areas, the thermal island effect, the level of income or the age of the population play in this investigation?

We have added an additional paragraph (Section 4.5) in the discussion to talk about the effects of air pollution, air conditioning, and the built environment.  Our study was at the population based, which is useful to inform health services (e.g. Queensland Ambulance Services) on a broad, state-wide wide policy level.

We have enquired about air quality data to the Department of Environment and Science. Currently, the air quality monitoring stations are mostly positioned along the coast, as pictured in the map below. There are no air quality monitoring stations in the Southwestern quadrant of the state. Therefore, we are unable to effectively approximate air quality at a state-wide level.

See attached for image

(https://apps.des.qld.gov.au/air-quality/)

We recognize the importance of understanding the dynamics of heat and health on an individual level. Collecting air conditioning data, green area data, age of buildings, etc. would likely require prospective analysis. We understand that there is work being undertaken in this space by other groups across Australia and look forward to seeing if we can link into this data.
